# Spatio-Temporal Changes in Vegetation in the Last Two Decades (2001–2020) in the Beijing–Tianjin–Hebei Region

Yuan Zou [1,2,3], Wei Chen [1,2,3,*], Siliang Li [1,2,3], Tiejun Wang [1,2,3], Le Yu [4], Min Xu [5], Ramesh P. Singh [6] and Cong-Qiang Liu [1,2,3]

1.  Institute of Surface-Earth System Science, School of Earth System Science,
    Tianjin University, Tianjin 300072, China
2.  Haihe Laboratory of Sustainable Chemical Transformations, Tianjin 300192, China
3.  Tianjin Bohai Rim Coastal Earth Critical Zone National Observation and Research Station,
    Tianjin 300072, China
4.  Ministry of Education Key Laboratory for Earth System Modeling, Department of Earth System Science,
    Tsinghua University, Beijing 100084, China
5.  State Key Laboratory of Remote Sensing Science, Aerospace Information Research Institute,
    Chinese Academy of Sciences, Beijing 100101, China
6.  School of Life and Environmental Sciences, Schmid College of Science and Technology, Chapman University,
    Orange, CA 92866, USA
*   Correspondence: chenwei19@tju.edu.cn

**Abstract:** In terrestrial ecosystems, vegetation is sensitive to climate change and human activities. Its spatial-temporal changes also affect the ecological and social environment. In this paper, we considered the Beijing–Tianjin–Hebei region to study the spatio-temporal vegetation patterns. The detailed analysis of a moderate-resolution imaging spectroradiometer (MODIS) data were carried out through the Google Earth Engine (GEE) platform. Our results show a slow and tortuous upward trend in the average leaf area index (LAI) in the study region for the periods 2001–2020. Specifically, Beijing had the highest LAI value, with an average of 1.64 over twenty years, followed by Hebei (1.30) and Tianjin (1.04). Among different vegetation types, forests had the highest normalized difference vegetation index (NDVI) with the range of 0.62–0.78, followed by shrubland (0.58–0.75), grassland (0.34–0.66), and cropland (0.38–0.54) over the years. Spatially, compared to the whole study area, index value in the northwestern part of the Beijing–Tianjin–Hebei region increased greatly in many areas, such as northwest Beijing, Chengde, and Zhangjiakou, indicating a significant ecological optimization. Meanwhile, there was ecological degradation in the middle and southeast regions, from Tangshan southeastward to Handan, crossing Tianjin, Langfang, the east part of Baoding, Shijiazhuang, and the west part of Cangzhou. Air temperature and precipitation were positively and significantly correlated with net primary production (NPP) and precipitation stood out as a key driver. Additionally, an intensification of the urbanization rate will negatively impact the vegetation NPP, with the shrubland and forest being affected most relative to the cropland.

**Keywords:** Beijing–Tianjin–Hebei region; normalized difference vegetation index (NDVI); average leaf area index (LAI); net primary production (NPP); multiple driving factors

## 1. Introduction

Global environmental change and sustainable development are two major challenges faced by human society [1]. Vegetation, as the main body of terrestrial ecosystems, is influenced by multiple factors, and in turn, its changes can affect climate change, surface albedo, roughness, carbon storage, and even social developing planning and processing [2]. In the case of China, as the largest developing country, its vegetation changes and human activities are active and have a great impact on the world. China accounts for only 6.6% of the global vegetated area but contributes 25% of the global net increase in leaf area,

with forests and croplands contributing 42% and 32%, respectively [3]. China's artificial impervious surface area now ranks first in the world, and together with the second-ranked United States, they occupy nearly 50% of the world's artificial impervious surface area [4].

In recent years, based on the remote sensing method, various vegetation indices were used as indicators on global, national, and regional scales to determine vegetation changes. The normalized difference vegetation index (NDVI) is an effective indicator to characterize the growth status and coverage of vegetation and monitor the ecological environment [5]. The leaf area index (LAI) is used to characterize vegetation canopy density and is also a key input parameter for terrestrial models [6,7]. Net primary productivity (NPP) is the direct basis for determining carbon sources/sinks in terrestrial ecosystems [8].

The measurement of vegetation dynamics includes in situ observation, the empirical formula method, optical instrument method, model estimation method, etc. [9,10]. To examine vegetation changes and their drivers, many approaches are used to qualitatively or quantitatively analyze the massive data. We used the Mann–Kendall test of trend significance for time series. This test is often used to analyze a hydrometeorological time series [11,12]. Sen's slope [13,14] and the slope of a linear regression equation [15] are also used to analyze the trend of time series data. Coefficient of variation can be used to compare the degree of dispersion of two sets of data without considering dimension [16]. The Pearson correlation coefficient reflects the degree of linear correlation between two variables [17]. Wang et al. [18] proposed that the geographical detector model (GDM) can not only test the stratified heterogeneity of a single variable, but also detect the possible causal relationship between two variables by testing the coupling of the spatial distribution of two variables.

As a direct factor influencing vegetation changes, climate factors attracted attention widely. Sun et al. [19,20] combined the LAI of China's ecologically fragile regions with climate factors and found temperature to be the dominant meteorological factor. Based on the boreal ecosystem productivity simulator (BEPS) ecological process mechanism model, Sun et al. [21] estimated the productivity of different ecosystems in the Beijing–Tianjin–Hebei region. The results disclose a significant positive correlation between precipitation and NPP ($P < 0.001$), the effect of precipitation being 3.95 times that of temperature. Zheng et al. [22] analyzed MODIS NDVI data and concluded that human activities and climate change contributed 42.35% and 57.65%, respectively, to the Delta NDVI on grassland in the Loess Plateau.

Among the factors of human activities, urbanization has a significant effect on the ecological and hydrological process [23], regional warming [24], vegetation phenology [25], and affects the temporal-spatial distribution patterns of vegetation profoundly. With rapid urbanization, human activities intensified, and a large amount of vegetated land was converted into urban land, affecting the functions of urban vegetation in water maintenance and climate regulation [26]. The process of urbanization will cause and aggravate ecological and environmental problems [27,28]. The relationship between urbanization development and vegetation growth will help to develop strategies for sustainable ecological development.

As the "Capital economic circle", the Beijing–Tianjin–Hebei region has vital economic and ecological meaning. Due to urbanization and industrialization, the development of this area is unbalanced, and the ecological environment is relatively fragile. Selecting the Beijing–Tianjin–Hebei region as the study area helps to understand the current situation and development trend of the vegetation resources through the analysis of the temporal-spatial change pattern and the driving factors, therefore serving for the improvement of regional ecological carrying capacity coordinate economic and social development. In the past 20 years, the area of vegetation improvement accounted for 88.09% and the area of degradation took up 11.91% in this region. Specifically, compared with the period of 2000–2009, the area of vegetation improvement decreased slightly, and the area of degradation increased during 2010–2019 [29].

We considered three vegetation indices based on vegetation cover, canopy structure, and vegetation productivity, combined with climate factors and urbanization rate data to

comprehensively analyze the forest, shrub, grassland, and cropland dynamics separately in the study region. The results will help to understand the changes and dynamics of various vegetation types in the Beijing–Tianjin–Hebei region.

## 2. Data and Materials

### 2.1. Study Area

The Beijing–Tianjin–Hebei region includes Beijing, Tianjin, and Hebei Province. It is located in North China at 113°27′–119°50′E, 36°05′–42°40′N, with a total area of 218,000 km² (Figure 1). The terrain is high in the northwest and low in the southeast, showing vertical climatic zones. The regional climate is temperate to warm-temperate and semi-humid to semi-arid continental monsoon climates [30]. Vegetation types mainly include deciduous broadleaf forests, mixed forests, closed shrublands, open shrublands, grassland, and cropland.

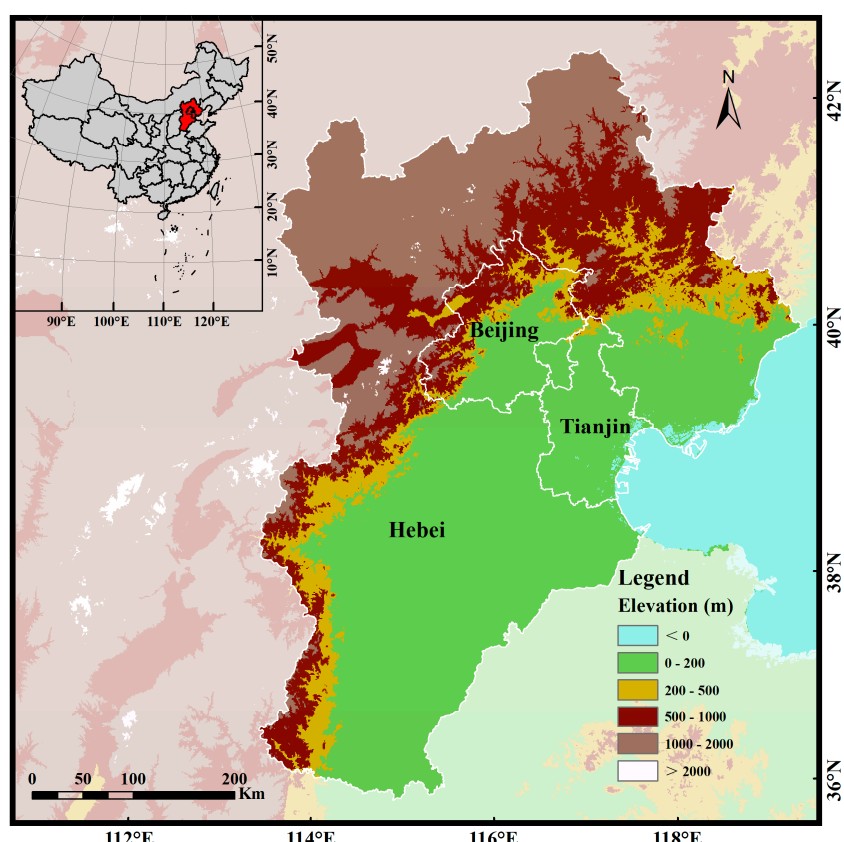

**Figure 1.** The location of the study area.

### 2.2. Datasets

The moderate-resolution imaging spectroradiometer (MODIS) is a sensor mounted on the Terra and Aqua satellites. It has 36 medium-resolution spectral bands, including 250 m (band 1–2), 500 m (band 3–7), and 1000 m (band 8–36). It is mainly used to obtain observational data of global biological and physical processes with a revisit cycle of 1–2 days.

All the vegetation indicators data are from the United States Geological Survey (USGS, Reston, VA, USA, https://www.usgs.gov, accessed on 24 February 2022). The LAI data come from the MOD15A2H dataset. The algorithm of this dataset is to select the best pixels from the Terra satellite sensors within 8 days to obtain a composite dataset with a period of 8 days and a resolution of 500 m. The NDVI data come from the MOD13Q1 dataset. The algorithm synthesis uses the images with less cloud, low viewing angle, and highest pixel value with a spatial resolution of 250 m. The NPP data comes from the MOD17A3HGF dataset, which is the improved MOD17. The annual NPP value is derived from the sum of

all 8-day net photosynthetic products in a given year and will be generated at the end of each year with a spatial resolution of 500 m.

The landcover classification criteria are based on the MODIS land cover type yearly global 500 m dataset and the annual land cover product of China (CLCD) [31]. For the MODIS dataset, band "LC_Type1" was selected as classification criteria, which is based on IGBP classification and divides the land use/land cover into 17 categories. The CLCD is Landsat-derived 30 m dataset. The land use is divided into cropland, forest, shrub, grassland, water, snow/ice, barren, impervious, and wetland. The annual average temperature, precipitation, and urbanization rates are from the official websites of statistics or metrology of each province/city.

### 2.3. Vegetation Indices

In this study, the normalized difference vegetation index (NDVI), leaf area index (LAI), and net primary productivity (NPP) are selected as the indicators of vegetation changes in the Beijing–Tianjin–Hebei region.

### 2.3.1. NDVI

NDVI is one of the important vegetation indices reflecting vegetation growth and nutritional information. NDVI can reflect the background information of the vegetation [32]. It is easily affected by the background value in sparse vegetation areas, and also easily reaches saturation in dense vegetation areas. The formula is:

$$NDVI = (B_{NIR} - B_R)/(B_{NIR} + B_R) \tag{1}$$

$B_{NIR}$ and $B_R$ stand for the reflection value in the near infra-red band and red band of images, respectively. The value range is [–1, 1]. A negative value indicates the landcover is rain, water, snow, etc., with high reflectivity; 0 indicates rock or bare soil; and a positive value indicates there is vegetation coverage. The larger the value, the greater the coverage.

### 2.3.2. LAI

LAI is the sum of the leaf area of all vegetation in a unit area of land. It is a comprehensive index that indicates the utilization of light energy and the canopy structure of vegetation. LAI is related to density, structure, biological characteristics of trees, and comprehensive environmental conditions. It can reflect the productivity of vegetation to a certain degree, moreover, effectively supplement the deficiency of NDVI in the high-density area [33].

### 2.3.3. NPP

NPP refers to the carbon absorbed by organisms through photosynthesis per unit time, excluding the plant's respiration. NPP can reflect the growth process of plant communities. It is the amount of organic carbon that is actually used for plant growth and also as an important indicator of the ability of plants to fix $CO_2$. NPP can be expressed as:

$$NPP = PAR \times FPAR \times \varepsilon - Ra \tag{2}$$

where PAR is photosynthetically active radiation; FPAR is the ratio of photosynthetically active radiation absorbed by vegetation; $\varepsilon$ is the realistic light energy utilization rate based on the concept of gross primary productivity (GPP), and Ra is vegetation autotrophic respiration.

### 2.4. Google Earth Engine Platform

Google Earth Engine (GEE) is a big data cloud computing platform that includes a variety of data sources commonly used in remote sensing analysis, allowing users to access high-quality data resources to process huge geographic data in batches. It is widely used in terrestrial studies. Based on the GEE platform, Thenkabail's group established a global 30 m resolution cropland extent production with a random forest machine learning

algorithm [34,35]. The World Resources Institute established a global forest monitoring website, Global Forest Watch, based on global forest dynamic monitoring realized by Hansen et al. [36]. Donchyts et al. [37] established a global surface water change observation website over the past 30 years. In addition to these, GEE has a wide range of applications in the information extraction of landcover [38], wetland landscape spatial pattern evolution [39], and degradation of forests under special events [40].

## 3. Methodology

### 3.1. Data Processing

This study, based on the GEE platform, set Beijing, Tianjin, and Hebei Province as the research area, selecting LAI, NDVI, and NPP from MODIS products between 2001 and 2020 to achieve long-term, large-scale batch processing. The workflow is shown in Figure 2. Taking Tianjin's 2001–2020 average LAI of the growing season as a case, the steps are as follows:

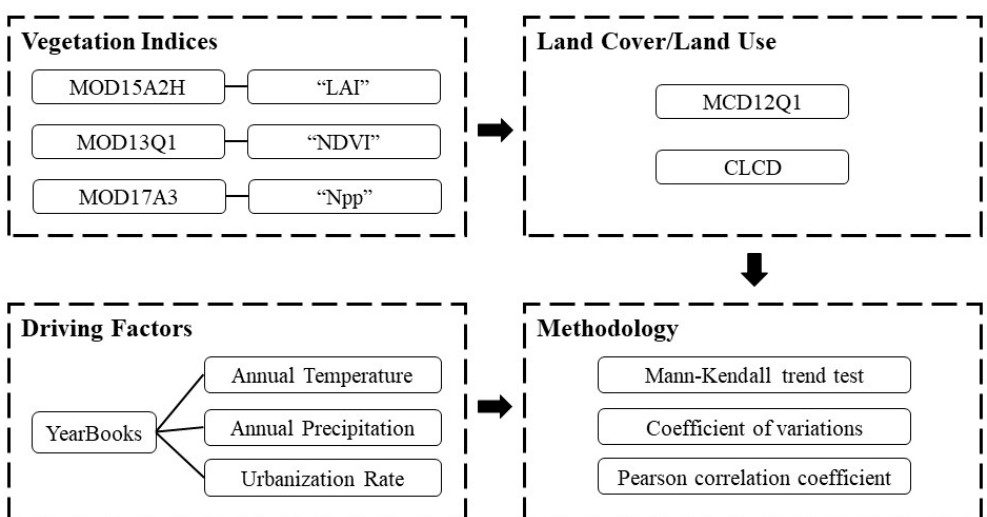

**Figure 2.** The workflow of this study.

Import the MOD15A2H dataset, MCD12Q1.006 products, and CLCD datasets into the GEE platform. After selecting bands, filter region, filter time period, quality control, and other pre-processing, we set the MODIS land cover classification value 1–5 as forest, 6–7 as shrubland, 8–10 as grassland, and 12 and 14 as cropland. Similarly, 1–4 represents cropland, forest, shrubland, and grassland, respectively, in CLCD.

Firstly, calculate the average value of the growing season LAI during 2001–2020. Secondly, mask them with MODIS landcover and CLCD, respectively. Obtain the patch images of forest, shrubland, grassland, and cropland in Tianjin. Finally, average the value in each category to obtain an average value of the growing season LAI of four landcover types. When the batch processing is completed on GEE, the average values of LAI in each area during the study period are calculated and analyzed. We add the layers in the form of visual images and export them to ArcGIS and other data processing software for further analysis [41].

### 3.2. Mann–Kendall Test

The Mann–Kendall test is used to analyze the changing trend of data, which can effectively distinguish whether a process is in natural fluctuation or there is a definite changing trend. It does not require the sample to obey a certain distribution, nor is it interfered with by a few outliers. It has a high degree of quantification and a wide detection range. It is suitable for sequential variables with a formula as:

$$slope = \frac{\sum_{i=1}^{n} x_i t_i - \frac{1}{n}\left(\sum_{i=1}^{n} x_i\right)\left(\sum_{i=1}^{n} t_i\right)}{\sum_{i=1}^{n} t_i^2 - \frac{1}{n}\left(\sum_{i=1}^{n} t_i\right)^2} \tag{3}$$

The *slope* represents the linear regression coefficient; $x_i$ is the vegetation index value of the year $i$; $t_i$ is the year $i$; and $n$ is the number of years under consideration [42,43]. When the *slope* is greater than 0, the change shows an increasing trend and vice versa.

### 3.3. Coefficient of Variations

Due to the different magnitude values of the three indices selected in this study, we choose the coefficient of variation to indicate the degree of the data fluctuation. It is a good way to compare the degree of dispersion between different types of parameters. The formula is:

$$CV = \frac{\sqrt{\sum_{i=1}^{n}(x_i - \overline{x})^2/(n-1)}}{\overline{x}} \qquad (4)$$

In the equation, CV is the coefficient of variation; $x_i$ is the vegetation index value of the year $i$; $\overline{x}$ is the average vegetation value within the study period; and $n$ is the number of years under consideration.

### 3.4. Pearson Correlation Coefficient

The Pearson correlation coefficient is defined as the ratio of the covariance and standard deviation between two variables and can reflect the degree of linear correlation between the two variables. The value range is $[-1, 1]$, the greater the absolute value of the coefficient, the stronger the correlation. The formula is:

$$r = \frac{\sum_{i=1}^{n}\left(X_i - \overline{X}\right)\left(Y_i - \overline{Y}\right)}{\sqrt{\sum_{i=1}^{n}\left(X_i - \overline{X}\right)^2}\sqrt{\sum_{i=1}^{n}\left(Y_i - \overline{Y}\right)^2}} \qquad (5)$$

Here $r$ is the correlation coefficient; $X_i$ and $Y_i$ are the values of the two variables (vegetation indicators and climate factors) in the year $i$, respectively; $\overline{X}$, $\overline{Y}$ are the average values of the two variables in the study period; and $n$ is the number of study years.

## 4. Results

We compared vegetation indicators value from MODIS IGBP landcover and CLCD to examine whether land use/landcover (LULC) maps with higher accuracy and resolution would perform better in this study. Take the growing season average LAI value in the Beijing–Tianjin–Hebei region for example, three-quarters of the vegetation types showed obviously different results under the two classification criteria of MODIS LULC and CLCD (Figure 3). Specifically, LAI values of forest and shrubland extracted by CLCD show an abnormal lower value than those from MODIS LULC and other researchers' results [44]. LAI value of grassland from MODIS is slightly higher than that from CLCD, while cropland has closer data under the two criteria. Generally, the LAI value quality defined by MODIS LULC is more suitable in this study.

The average LAI value of the whole area defined by both criteria is the same, indicating that the difference in the LAI value of each vegetation type comes only from different classification criteria. It shows although CLCD has a higher spatial resolution, the MODIS LULC is more consistent with other vegetation indices data resources, we will use the data defined by MODIS IGBP landcover for the following analysis.

In terms of LAI and NDVI values, Beijing outperforms Hebei, followed by Tianjin. For NPP, the results show that the value in Beijing and Hebei are close, with Tianjin remaining the lowest, relatively. The MODIS IGBP landcover change in the study region has shown in Figure 4. Overall, deciduous broadleaved forests and mixed forests are the dominant forest types in Beijing. Tianjin was dominated by deciduous broadleaf forests before 2015, and then sizable mixed forests grew or were planted. Hebei's forest species are more abundant, including evergreen needleleaf forests, deciduous needleleaf, deciduous broadleaf forests, and mixed forests. The shrubs in Beijing, Tianjin, and early Hebei are major closed shrublands. However, since 2016, sizable open shrublands appeared in Hebei.

All the grass types (woody savannas, savannas, and grassland) can be seen in the Beijing–Tianjin–Hebei region. The functions of cropland in Beijing and Tianjin are relatively simple, while in Hebei, there are large areas where cropland and natural vegetation are mixed.

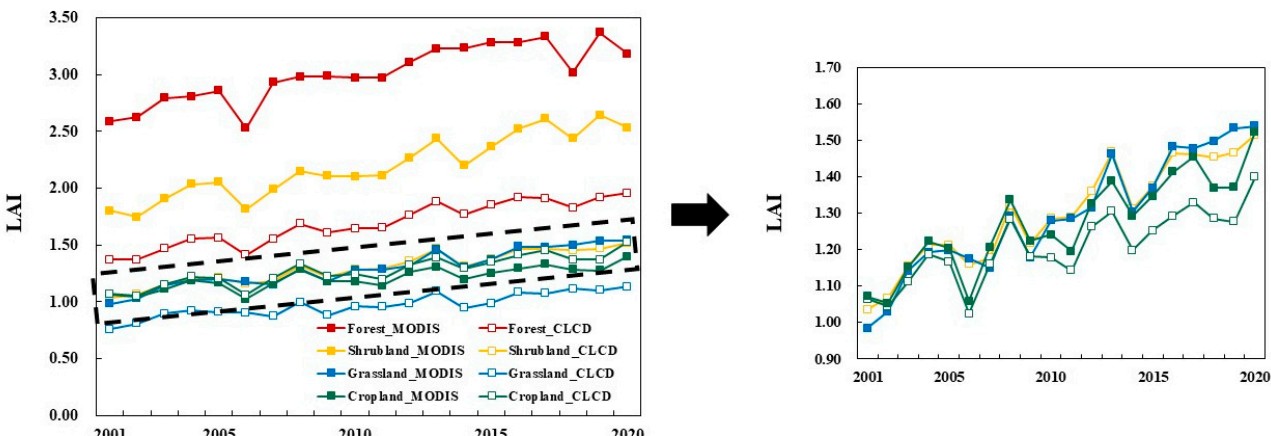

**Figure 3.** The growing season average LAI of four landcover types in the Beijing–Tianjin—Hebei region using two classification criteria of MODIS IGBP and CLCD.

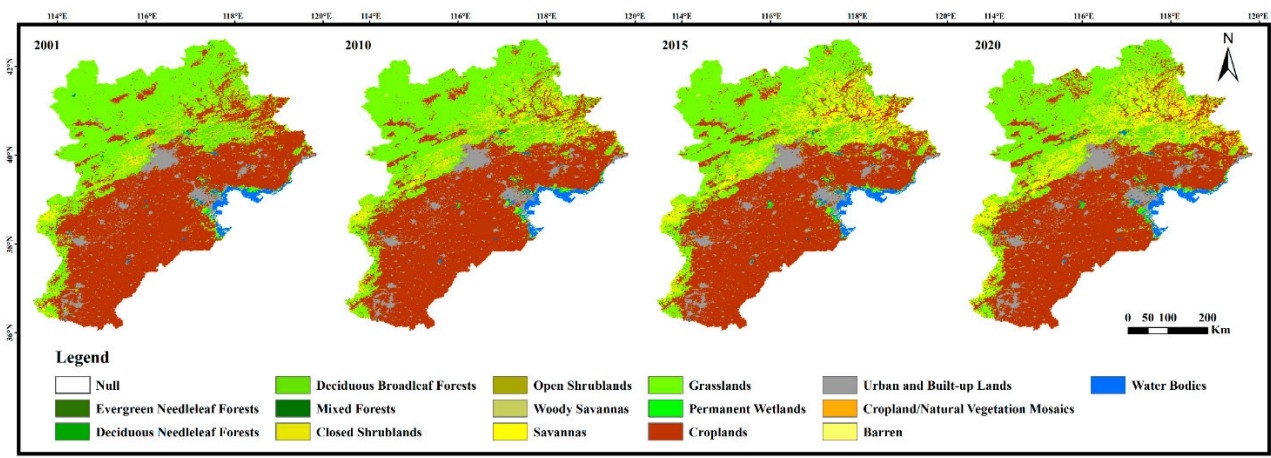

**Figure 4.** The MODIS IGBP landcover of Beijing–Tianjin–Hebei in 2001, 2010, 2015, and 2020.

### 4.1. NDVI

Figure 5a shows the overall trend of the average value of growing season NDVI in the Beijing–Tianjin–Hebei region during 2001–2020. Over the past 20 years, the average NDVI value increased from 0.43 to 0.54 with a mean value of 0.48. During 2001–2011, the NDVI value fluctuated in the study area. The value reached the peak in 2004 and 2008, with values of 0.49 and 0.50, respectively, and then dropped to a trough of 0.43 in 2006. The change in NDVI value was only 0.01/10a in the first ten years. After 2010, accompanied by two sudden increases in 2011–2012 and 2015–2016, the region's NDVI value increased intensely. The NDVI change during 2011–2020 was 0.09, which was about 9 times that during 2001–2010.

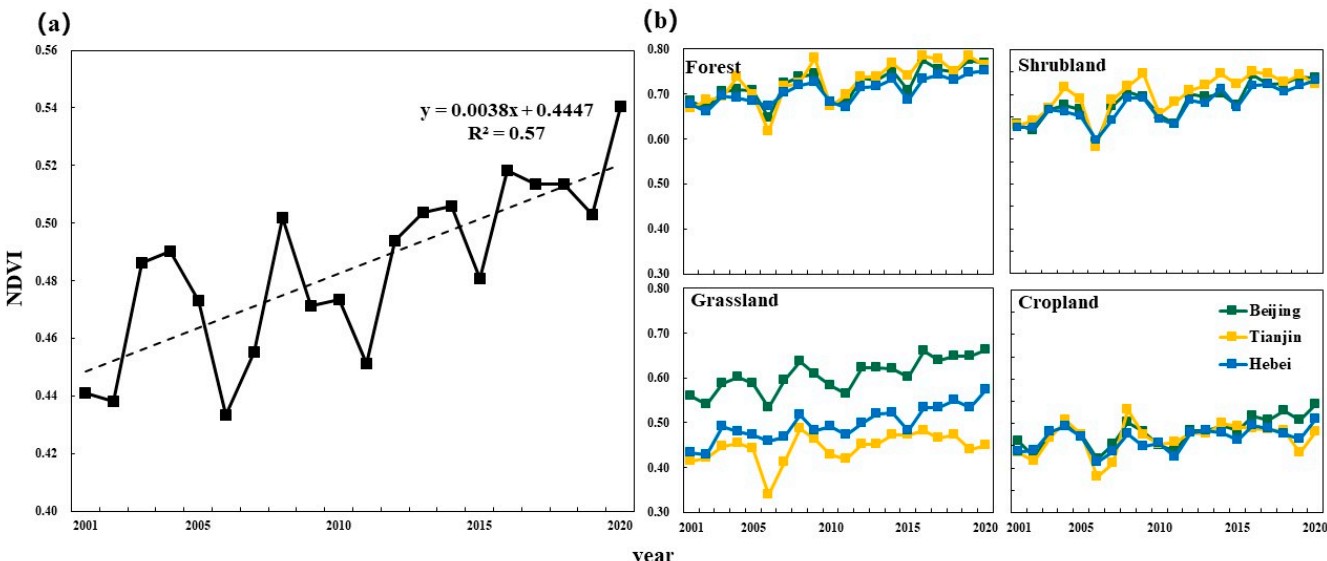

**Figure 5.** Changes in the average value of growing season NDVI in the Beijing–Tianjin–Hebei region, ((**a**) average value of growing season and (**b**) growing season average value of four landcover types).

The temporal change of four vegetation types in Beijing, Tianjin, and Hebei are consistent with the overall trend (Figure 5b). The vegetation types under consideration include forests showing the highest NDVI, followed by shrubland, grassland, and cropland, respectively. With similar performance and the close value of forest, shrubland, and cropland in NDVI values, the grassland shows an obvious heterogeneity, showing the highest (0.61) in Beijing, Hebei as the second (0.50), and Tianjin shows the lowest (0.44) value.

### 4.2. LAI

The average change of vegetation growing season LAI value in the Beijing–Tianjin–Hebei region is shown in Figure 6. Although more fluctuate, the LAI of the study area shows a similar pattern to the temporal change of NDVI, with Beijing outperforming Hebei and Tianjin. LAI rose from 1.08 to 1.53 with an increase of 0.45 during 2001–2020. In terms of landcover types, the forest and shrubland have similar performance, with an average value of 2.92–3.10 and 2.16–2.36, respectively. The LAI value of grassland in Beijing (1.68) was obviously higher than that of Hebei (1.26) and Tianjin (1.03). As for cropland, the value was relatively higher in Hebei (1.22), while Beijing and Tianjin are relatively lower with 1.12 and 1.13, respectively. The lowest LAI is distributed in the northwest and eastern parts of the whole region. The value in the northeast and a small part of the southwest are obviously higher, along with a middle level in the south-central and southeastern regions.

### 4.3. NPP

As shown in Figure 7, the vegetation NPP in Beijing and Hebei is close, with the annual values of 318.28 g C/m$^2$ and 320.24 g C/m$^2$, respectively, which are obviously higher than the value of 275.33 g C/m$^2$ in Tianjin. In 2001, Beijing had the lowest NPP value of 176.77 g C/m$^2$. After 20 years of rapid growth, Beijing had the highest NPP value of 386.57 g C/m$^2$ at 2020 among the whole study area. Generally, NPP has a similar trend as LAI and NDVI. However, in some years (e.g., 2007, 2011, and 2015), NPP showed an opposite trend to the other two indicators. For instance, in 2007, NDVI and LAI started to increase compared to last year, while NPP still decreased and get to the trough at that time.

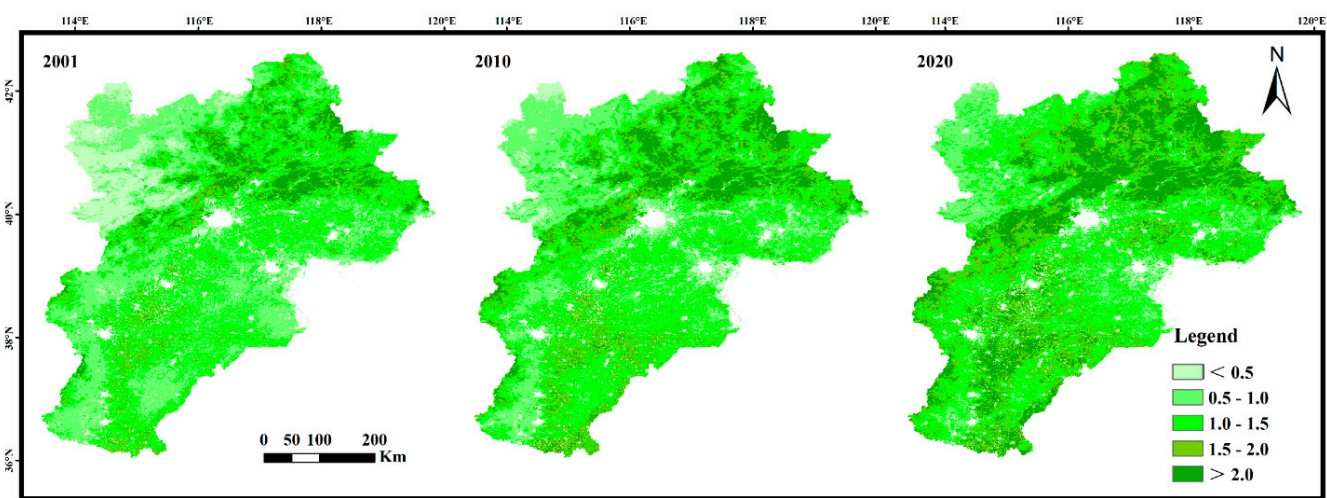

**Figure 6.** Spatial distribution of the average growing season LAI value in 2001, 2010, and 2020.

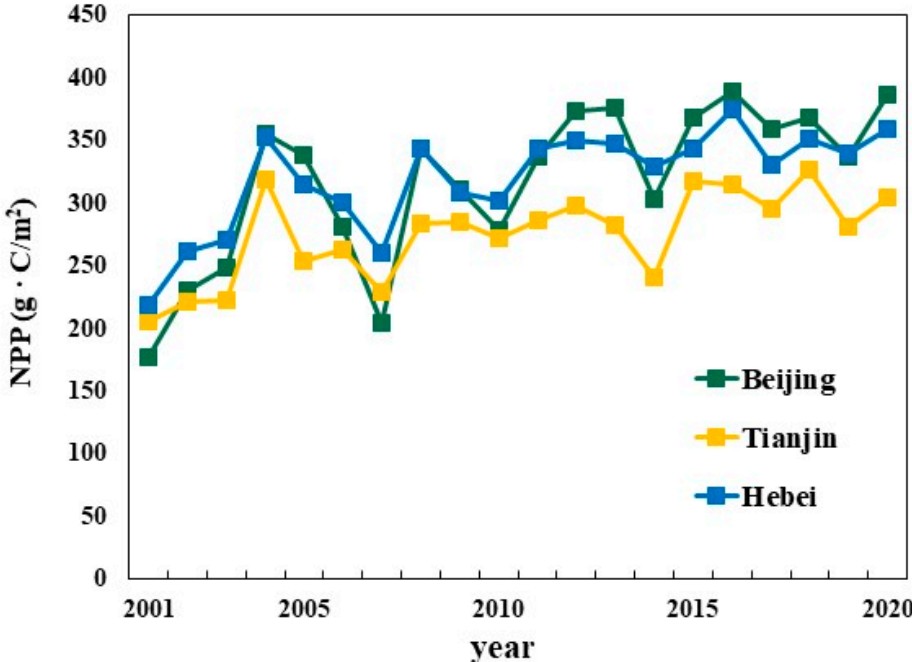

**Figure 7.** The annual value of NPP in Beijing, Tianjin, and Hebei.

### 4.4. Spatio-Temporal Characteristics of Indices

The vegetation indices in Beijing, Tianjin, and Hebei showed increasing trends during the study period, indicating the overall optimization of the ecological environment in the study area of this region. Zhang et al. [45] believed that the main reason for this trend was the implementation of the Grain for Green Project and the building of many safety barriers in Beijing, Tianjin, and Hebei in 2000. In comparison, Hebei has more cropland and Beijing–Tianjin have more built up. This shows that Hebei Province has a relatively wide area, which means it should probably take more responsibility for crop production in order to supply the others due to the Beijing–Tianjin–Hebei integration process.

The result of indices analyzed by the Mann–Kendall trend test and coefficient of variation are shown in Table 1. We take a two-sided test with $p < 0.025$ as significant, and the general trend of the Beijing–Tianjin–Hebei region shows an optimization variation with an M-K *slope* greater than zero. Over the twenty years, the overall situation of the vegetation LAI and NDVI in the study area remained relatively stable, with CV less than 0.15, while NPP changes are relatively unstable, especially for forest and shrubland in Tianjin.

The spatial variation in vegetation indices in the Beijing–Tianjin–Hebei region during 2001–2020 is shown in Figure 8. The data hierarchical display is manually set to "0" points based on Jenks's method. Among them, the NPP increased the most, with the increase being most significant in the north and west part of the research area, while the eastern sporadic area decreased slightly. Despite increasing less, the NDVI and LAI did also show an overall increasing trend. Their spatial variation characteristics are consistent with NPP, showing an optimization in the northwest and a certain degree of ecological degradation in the middle and southeast regions.

**Table 1.** M-K trend and CV of indices in the Beijing-Tianjin-Hebei region.

|  |  | M-K | | | CV | | |
|---|---|---|---|---|---|---|---|
|  |  | **Beijing** | **Tianjin** | **Hebei** | **Beijing** | **Tianjin** | **Hebei** |
| **LAI** | Overall | 0.0302 | 0.0098 | 0.0212 | 0.1178 | 0.0747 | 0.1033 |
|  | Forest | 0.0404 | 0.0454 | 0.0362 | 0.0966 | 0.1076 | 0.0805 |
|  | Shrubland | 0.0457 | 0.0451 | 0.0416 | 0.1377 | 0.1287 | 0.1170 |
|  | Grassland | 0.0324 | 0.0086 | 0.0263 | 0.1244 | 0.0710 | 0.1292 |
|  | Cropland | 0.0115 | 0.0133 | 0.0143 | 0.0795 | 0.0864 | 0.0835 |
| **NDVI** | Overall | 0.0057 | 0.0025 | 0.0037 | 0.0728 | 0.0791 | 0.0605 |
|  | Forest | 0.0045 | 0.0054 | 0.0037 | 0.0508 | 0.0624 | 0.0397 |
|  | Shrubland | 0.0054 | 0.0053 | 0.0051 | 0.0609 | 0.0645 | 0.0566 |
|  | Grassland | 0.0051 | 0.0025 | 0.0054 | 0.0629 | 0.0751 | 0.0760 |
|  | Cropland | 0.0039 | 0.0020 | 0.0021 | 0.0686 | 0.0774 | 0.0550 |
| **NPP** | Overall | 7.6141 | 4.1038 | 4.9789 | 0.1971 | 0.1304 | 0.1260 |
|  | Forest | 9.0827 | 8.1475 | 8.7888 | 0.2611 | 0.4419 | 0.1880 |
|  | Shrubland | 7.6701 | 6.9733 | 4.4660 | 0.2429 | 0.3833 | 0.1689 |
|  | Grassland | 8.5222 | 3.9385 | 6.8123 | 0.2089 | 0.1311 | 0.1423 |
|  | Cropland | 6.6101 | 4.4818 | 3.6477 | 0.1667 | 0.1367 | 0.1146 |

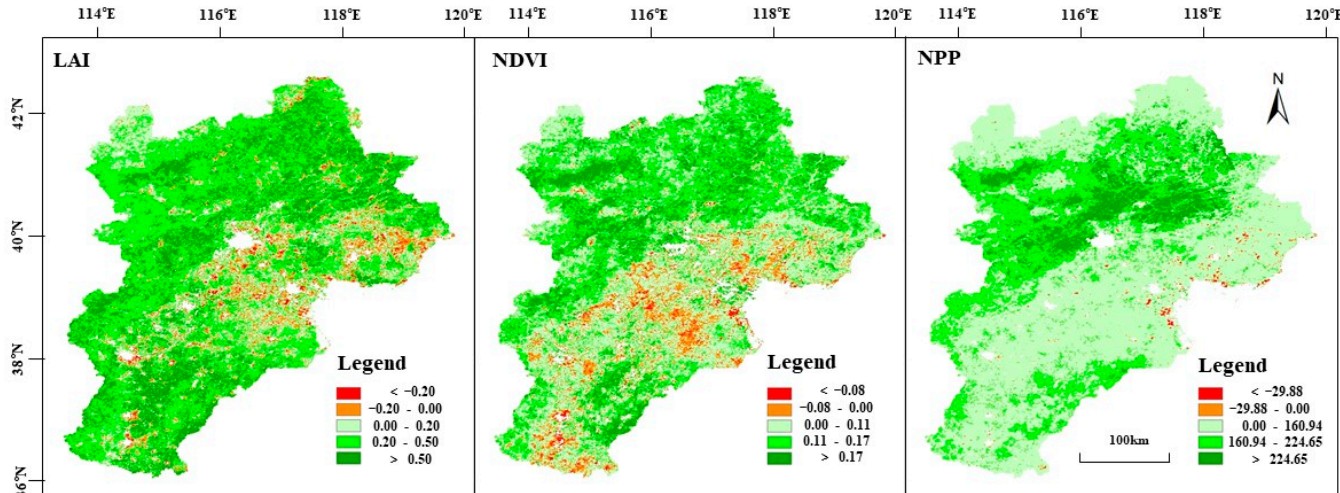

**Figure 8.** Spatial variation in Beijing–Tianjin–Hebei ecological parameters from 2001 to 2020.

The regional averages of the four indices in Beijing sequentially decreased by forest-shrubland, grassland and cropland, indicating that cropland contributed the least to the overall ecosystem. In Tianjin, cropland contributes most to the whole region, because the regional average values are between cropland and the other three types. The regional average value is close to that of cropland, showing that Hebei's ecological system is mainly determined by cropland, though the other three vegetation types have a positive influence on the regional value.

### 4.5. Climate Factors and Vegetation

The Pearson correlation coefficient of temperature, precipitation, and NPP are shown in Figure 9. Generally, temperature and precipitation are positively correlated with the NPP, indicating that the optimum temperature and precipitation required in this area are still higher than current conditions. To a certain extent, the increase in precipitation will not inhibit the production ability of vegetation. The significance probability between NPP and temperature shows very low statistical meaning, opposite, NPP with precipitation shows high significance.

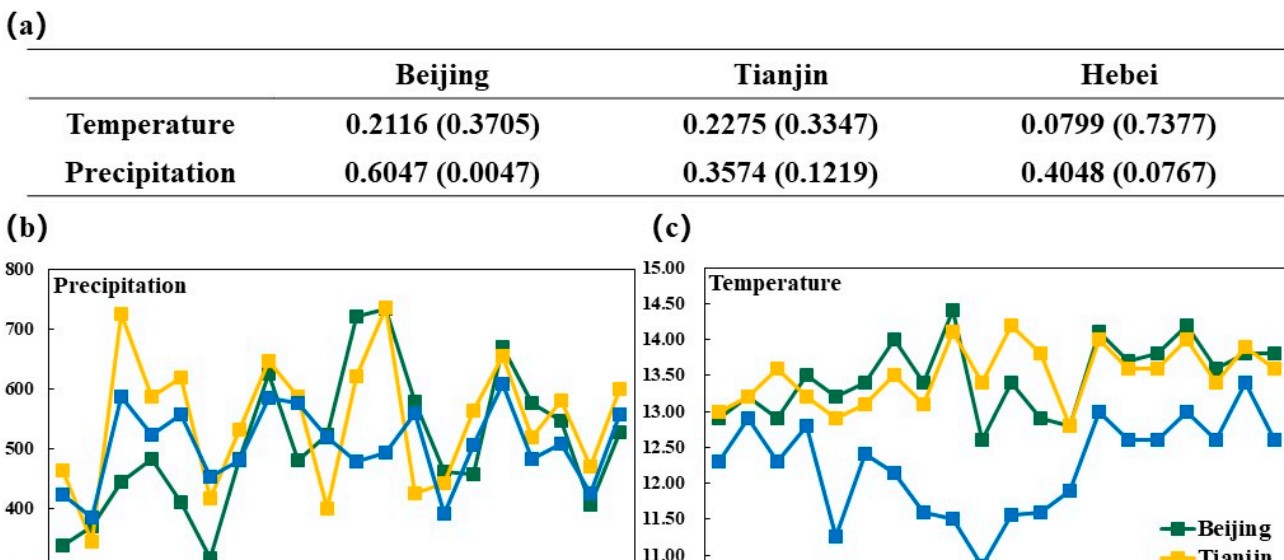

**(a)**

|  | Beijing | Tianjin | Hebei |
|---|---|---|---|
| **Temperature** | 0.2116 (0.3705) | 0.2275 (0.3347) | 0.0799 (0.7377) |
| **Precipitation** | 0.6047 (0.0047) | 0.3574 (0.1219) | 0.4048 (0.0767) |

**Figure 9.** (**a**) Pearson correlation coefficient and the significance probability between climate factors and NPP; (**b**) regional annual temperature change trend; and (**c**) regional annual precipitation change trend.

### 4.6. Urbanization Rate and Vegetation

A comparison of the relationship between the vegetation coverage of different land-cover types and the urbanization rate is shown in Figure 10. The urbanization rate in Tianjin and Beijing shows an increase from 2005 to 2006, at the same time, the vegetation productivity decreased significantly. The increase in the urbanization rate shows a pronounced change in the NPP fluctuation.

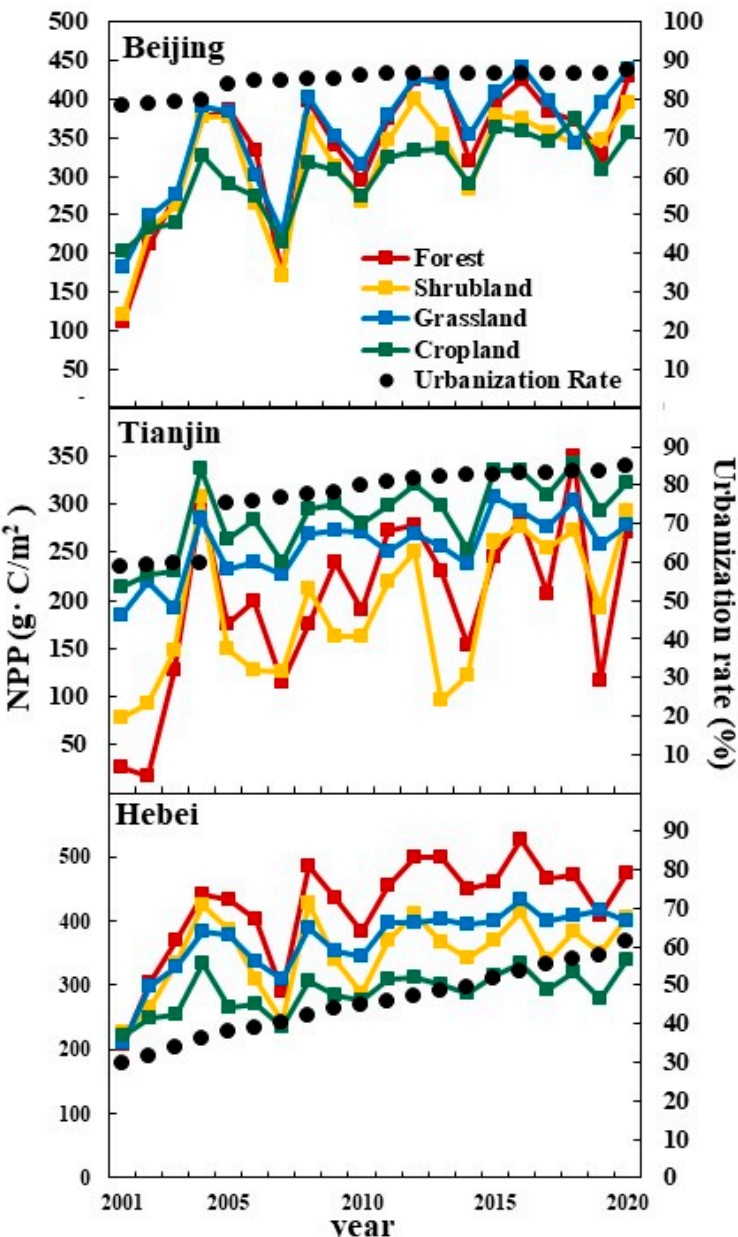

**Figure 10.** The relationship between the vegetation productivity of different landcover types and the urbanization rate.

## 5. Discussion

### 5.1. The Influence of Climate Factors on the Vegetation

The vegetation NPP is determined by plant photosynthesis and respiration, therefore, it can represent the carbon sequestration capacity of vegetation. Combining Figures 5a and 7, it can be seen that the performance of NPP is opposite to LAI and NDVI in certain years, such as 2007, 2011, and 2015. Moreover, The NPP is more consistent with temperature during 2001–2008 and has more correlation with precipitation in later years. To be more specific, we can see there is a strong positive correlation between precipitation and NPP in Beijing and a relatively significant correlation in Tianjin and Hebei. That is, the main influencing climate factor is precipitation, while temperature has a relatively weak influence, which is consistent with Piao [46]. As LAI and NDVI are more related to photosynthesis, we assume that the impact of climate on the vegetation productivity in this area is mainly realized through respiration.

*5.2. The Influence of Urbanization on the Vegetation*

Although part of the region promoted vegetation cover rate because of multiple afforestation programs and construction of wetlands/greenbelts, a large area of vegetation would be transformed into construction land with the process of urbanization. In this situation, urbanization will lead to the reduction in vegetation area, vegetation productivity degradation, ecological environment deterioration, and other adverse consequences.

From the perspective of the urbanization rate, Beijing started with a relatively higher urbanization level in 2001 from 78.06%, which is much higher than Tianjin during that time (58.56%), as well as Hebei (29.43%). With the coordinated development of Beijing, Tianjin, and Hebei, the urbanization rate of Tianjin increased intensely (44.6%) over the past 20 years and almost reached the level of Beijing. Although the level of urbanization in Hebei is still lower than that of Beijing and Tianjin, it increased by as high as 107.78% over the past 20 years.

Shrubland and forest are most affected during this process, followed by grassland and farmland, indicating that the popularization and urbanization process in this period lead to massive woodland and shrub loss. Since 2006, urbanization is no longer the main factor affecting vegetation production capacity, as the growth rate of urbanization in the Beijing–Tianjin–Hebei region since remained relatively low.

*5.3. Limitations and Future Work*

For the analysis of vegetation changes year by year, we used the average value of the growing season. In general, April to October every year is considered to be the growing season. However, with the extension of the growing season, the month selection may be different [47]. In our future study, we will consider the growing season to find whether it affects our results.

Secondly, the Beijing–Tianjin–Hebei region is a typical research area for studying vegetation changes and the influence of urbanization. For the next step, we are planning to expand the research area to discuss whether the regions surrounding can affect each other and obtain more interesting results in larger scale.

## 6. Conclusions

In this study, we illustrate the use of the GEE platform in conjunction with MODIS data to analyze regional vegetation spatio-temporal patterns, herein characterized by NDVI, LAI, and NPP. The Mann–Kendall trend test, coefficient of variation, and Pearson correlation coefficient were used to determine the trend change in three vegetation indices and the relationship between urbanization, climate factors, and the indices in the Beijing–Tianjin–Hebei region during 2001–2020. This study can provide scientific support for analyzing the impact of vegetation change on the ecological environment in the Beijing–Tianjin–Hebei region, and the following conclusions were drawn:

First, over 20 years, the average values of vegetation LAI, NDVI, and NPP in the Beijing–Tianjin–Hebei region showed a slow and tortuous upward trend. The vegetation types under consideration include forests, which stood out with the highest NDVI, followed by shrubland, grassland, and cropland, successively. Spatially, the northwestern region increased greatly, with significant ecological optimization. In contrast, the middle and southeast regions suggest an ecological degradation.

Then, generally, there is a positive correlation between temperature, precipitation, and the NPP, with precipitation standing out as the key driver. In the context of global warming, the increase in temperature will not inhibit the growth of vegetation in the study area temporarily. Moreover, the effect of climate on vegetation mainly depends on the respiration of vegetation. At the same time, the influence of climate factors in Beijing, Tianjin, and Hebei decrease in turn, which correspondingly indicates that the impact of human activities increases successively.

Lastly, with the process of urbanization, a large area of vegetation was transformed into urban construction land, which affects the vegetation's ecological functions and services.

When the urbanization rate increases suddenly, shrubs and forests are the most affected, while cropland is the least affected.

**Author Contributions:** Conceptualization, W.C. and Y.Z.; methodology, W.C. and Y.Z.; software, Y.Z.; validation, Y.Z.; formal analysis, Y.Z.; investigation, Y.Z.; resources, W.C. and Y.Z.; data curation, Y.Z.; writing—original draft preparation, Y.Z.; writing—review and editing, Y.Z., W.C., S.L., T.W., L.Y., M.X. and R.P.S.; visualization, Y.Z.; supervision, W.C. and C.-Q.L.; project administration, W.C.; funding acquisition, W.C. All authors have read and agreed to the published version of the manuscript.

**Funding:** This research was funded by the National Natural Science Foundation of China (Grant No. 42077168) and the Open Fund of State Key Laboratory of Remote Sensing Science (Grant No. OFSLRSS202105).

**Data Availability Statement:** Publicly available datasets were analyzed in this study. This data can be found here: [https://www.usgs.gov; http://tjj.beijing.gov.cn/; https://stats.tj.gov.cn/; http://tjj.hebei.gov.cn/ (accessed on 15 June 2022)].

**Acknowledgments:** We thank the Haihe Laboratory of Sustainable Chemical Transformations for financial support.

**Conflicts of Interest:** The authors declare no conflict of interest.

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
