# Peer review of "Spatio-Temporal Changes in Vegetation in the Last Two Decades (2001–2020) in the Beijing–Tianjin–Hebei Region"

_remotesensing, doi:10.3390/rs14163958_

Round 1
Reviewer 1 Report
Review report for "Spatio-temporal changes in vegetation in the last two decades (2001-2020) in the Beijing-Tianjin-Hebei region" by Yuan Zou and co-workers
In this study, the authors considered use three vegetation indices based on vegetation cover, canopy structure, and vegetation productivity, in combination with climate factors and urbanization rate data to analyze the forest, shrub, grassland, and cropland dynamics in the Beijing-Tianjin-Hebei region based on two decades (2001-2020) of data.
The three indices under consideration iclude the Normalized Difference Vegetation Index (NDVI), the Leaf Area Index (LAI), and the Net Primary Productivity (NPP) selected as indicators of vegetation changes in the Beijing-Tianjin-Hebei region.
In terms of LAI and NDVI values, Beijing > Hebei > Tianjin; for the NPP, Beijing ≈ Hebei > Tianjin. Overall
GENERAL APPRAISAL
Analyses of this kind are important since the results may help to understand the changes and dynamics of various vegetation types in regions of interest.
The MS is relatively well written, but I encourage the authors to get the MS checked for English language.
SPECIFIC COMMEMTS
Most of my comments are minor
L20 Insert a space between "the" and "average"
L21: Result show a slow and tortuous upward trend in average leaf Area index (LAI) in the study region between 2001 and 2020.
L25-28: Spatially, compared to the whole study area, index values in the northwestern part of the Beijing-Tianjin-Hebei region increased greatly in many areas such as northwest Beijing, Chengde, and Zhangjiakou [near there?], indicating a significant ecological optimization.
L30: Temperature and precipitation were positively and significantly correlated with net primary production (NPP) and precipitation stood out as key driver.
L32-34: Additionally, an intensification of the urbanization rate will negatively impact the vegetation NPP, with the shrubland and forest being affected most relative to cropland.
L44: The sentence "Vegetation changes ... in the world" is unclear
L51:... are used
L60: the various relationships
L60: We used the Mann-Kendall test of trend significance for time series. This test is often used to analyze hydrometeorological time series [11,12].
l63: ... are also used.
l72: ... and found temperature to be the dominant meteorological factor.
l75: The results disclosed a significant positive correlation between precipitation and NPP, the effect of precipitation being 3.95 times that of temperature.
L77: Zheng at a. [22]] analyzed MODIS NDVI data and concluded that human....
L92 replace "It" by "it"
L122: There is no verb in this sentence, please fix.
L194 and L198: Insert a space between 'of" and 'LAI"
L212: x_i is the value of the year i [do you mean the value of something in year i, and if so, the value of what?], t_i is the year i.
Fig4: No apparent change in landcover can be detected visually by looking at Fig. 4. What message is Fig 4 intended to convey?
Author Response
Many thanks for sparing your valuable time and providing your valuable comments. We have revised our manuscript in the light of your comments, and a pointwise reply is also attached. We hope that our revised manuscript will be acceptable to this learned Referee.

Reviewer 2 Report
Extensive editing of English language and style are required. Comments in file attached.

Author Response
Many thanks for sparing your valuable time and providing comments/suggestions to improve our manuscript. We have now revised our manuscript in view of the comments/suggestions. Following your comments now we have checked English language and grammar of the manuscript with the help of a native English speaker. We hope that our revised manuscript will be acceptable to this learned Referee. Pointwise reply to the comments/suggestions made by Referee #2 is attached.

Round 2
Reviewer 2 Report
Accept in present form
This manuscript is a resubmission of an earlier submission. The following is a list of the peer review reports and author responses from that submission.
Round 1
Reviewer 1 Report
The manuscript “Spatio‐temporal changes in vegetation in the last two decades (2001‐2020) in the Beijing‐Tianjin‐Hebei region” used MODIS NIVI, LAI, and NPP to examine the characteristics of vegetation change patterns. Overall, there are nothing new in methodology, the structure is not well organized, and there are many English writing problems, it needs intensively revision.
General comments:
- The authors used three vegetation indicators to examine the vegetation changes. We know although there are different indicators, but highly correlated. Three parameters show different change trends, should address why.
- In literature review, authors did not summarize currently used approaches for examining vegetation changes.
- The purpose of this research is to analyze the spatial and temporal vegetation change, but did not see much spatial analysis. Only analyzed Beijing, Tianjin and Hebei regions. Is Figure 7 the M-k spatial distribution map? Please specify “spatial variation”. How big the m-k values are considered as significant?
- NDVI and LAI are significantly influenced by seasons, the averages of whole years are not appropriate for comparison of different regions. Averages of growing season are optimal options.
- Because Beijing, Tianjin, especially Hebei cover huge areas, the climate conditions vary over the large regions, thus, the Pearson correlation between vegetation parameters and precipitation / temperature over large scale cannot attribute the vegetation changes to climate at finer scale. Actually, the correlation coefficient or significant level map at pixel level also can be created, then you can see which areas are more affected by temperature or precipitation.
- The analyses of vegetation change factors should put in result section, not discussion.
- Many English grammar errors. Needs English professionals to check.
Specific comments:
- Line 21: not clear. “average of 1.2”, is it the average of 20 years over Beijing area, or a specific year? Why NDVI is a range, such as 0.44-0.68, not one value? Many numbers in the paper in other places have the same problems. All these numbers should be clarified.
- Line 44, Line 57 reference 3, 11 are too old.
- Line 48-52, 59-67, 78-84, sentences are too long, and hard to understand.
- Line 72 74, 78 “urban agglomeration vegetation”, “lead to generation …problems” “construction of urbanization and in..” are awkward.
- Section 2.3 datasets could put after study area. Introduction of dataset should be concise.
- Two types of LULC datasets were generated from different remote sensing data and methods, and the LULC systems are not same. Because of spatial resolution of CLCD is different from MODIS datasets used here, how to resample?
- Section 2.4 too many irrelevant literatures.
- Both Section 2.5 and 3 statistical analysis belong to methodology and should put together, and describe them in detail.
- Equation (2), (3), (4) are not clear, should be typed in, not copied as pictures. Because the paper is submitted to a professional journal, common terms such CV and r do need so detailed description.
- Section Results: the impact of two LULC on the LAI result, I did not see any meaning of comparison. Because two LULC datasets have different spatial resolution, and accuracy. Why MODIS LULC is more suitable here? The differences mostly attribute to the accuracy of classification.
- Line 223-231. What is this paragraph used for? Explain what?
- For each vegetation parameter, should analyze on study region, each specific region from both spatial and temporal perspectives, or keep consistent.
- When analyze vegetation change trends, should specify whether these changes are significant or randomly.
- How to define urbanization rate?
Reviewer 2 Report
Reviewer's Report for 'Spatio‐temporal changes in vegetation in the last two decades (2001‐2020) in the Beijing‐Tianjin‐Hebei region"
By Zou and co-workers
This study illustrate the use combine Google Earth Engine (GEE) platform in conjunction with Moderate‐resolution Imaging Spectroradiometer (MODIS) data to analyze regional vegetation spatio-temporal patterns herein characterized by three indices namely, the Normalized Difference Vegetation Index (NDVI), the average Leaf Area Index (LAI) and the net primary production (NPP). The paper includes a real-world application to two decades (2001‐2020) of data from the Beijing‐Tianjin‐Hebei region.
The average Leaf Area Index (LAI) in the region has shown a slow upward trend over the study period, with Beijing outperforming Hebei and Tianjin. The vegetation types under consideration include forests which stood out with the highest NDVI followed by shrubland, grassland, and cropland, respectively.
The index values reveal a remarkable spatial heterogeneity across the study area, with an increasing trend in the northwestern part of the Beijing‐Tianjin‐Hebei region indicating a significant ecological optimization, in contrast with the middle and southeast regions, suggesting an ecological regression.
Temperature and precipitation appeared to be positively correlated with NPP with precipitation standing out as the key driver. Fluctuation in vegetation NPP increases with the urbanization rate, with shrubland and forest being affected than cropland.
GENERAL APPRAISAL
This is an interesting topic, but the writing of the manuscript leaves a lot to be desired. In particular, statistical formulae are used generically. For some equations, the variables or their indices are not explained. The interpretation of the results is another area requiring improvement . Below I provide detailed comments.
SPECIFIC COMMENTS
[1] L66: On L66 one can read "... the effect of precipitation on NPP is 3.95 times that of temperature." Were the two variables initially standardized? Does "effect" here refer to the correlation coefficient?
It is worth keeping in mind that correlation does not imply causation.
[2] L68: How was urbanization quantified?
[3] L78: Construction of urbanization or just urbanization?
[4] L98: Please write km^2 instead of km2.
[5] L111-112: The sentence "It is easily... in dense vegetation areas" is incomplete.
[6] L115: What do the indices NIR and R stand for in equation (1)?
[7] L125-129: Please provide a description of how NPP computed.
[8] L187: Please indicate the response variable and the explanatory variable of the regression referred to on L187.
[9] L188-189: The statement "x_i is the value of the year i; t_i is the year i" is confusing. What is the difference between i and t_i as they both appear to indicate the year?
[10] L189: I would suggest replacing "number of concerned years" by "number of years under consideration".
[11] L203: Equation (4) is a generic formula. What are the two variables in X and Y in the context of this study?
[12] L259: "significantly higher" is a statistical statement. However the authors do not provide any indication on the test statistic used.
[13] L281-282: The sentence "The vegetation indices in Beijing, Tianjin, and Hebei ... indicating in this region." is incomplete: indicating in this region what??
[14] L283: Implementation of the Grain for Green project.
[15] Please provide a short description of the Man-Kendall trend test and the interpretation of ensuing results.
Reviewer 3 Report
Regarding the MS ID: remotesensing-1718602
The paper seems really nice and matches the scope of the Journal. I would accept the paper after addressing the comments:
1- The range of the NDVI changes during 20 years is not that large, and it can be related to mostly forest, is it ok to sum up with other land uses.?
2 -quality of the figures is low, please enhance the figure qualitteis,
3 - The discussion section should be improved.